# MaskMamba: A Hybrid Mamba-Transformer Model for Masked Image Generation

## Abstract

Image generation models have encountered challenges related to scalability and quadratic complexity, primarily due to the reliance on Transformer-based backbones. In this study, we introduce MaskMamba, a novel hybrid model that integrates Mamba and Transformer architectures, utilizing Masked Image Modeling for non-autoregressive image synthesis. We meticulously redesign the bidirectional Mamba architecture by implementing two key modifications: (1) replacing causal convolutions with standard convolutions to better capture global context, and (2) utilizing concatenation instead of multiplication, which significantly boosts performance while accelerating inference speed. Additionally, we explore various hybrid schemes of MaskMamba, including both serial and grouped parallel arrangements. Furthermore, we incorporate an in-context condition that allows our model to perform both class-to-image and text-to-image generation tasks. Our MaskMamba outperforms Mamba-based and Transformer-based models in generation quality. Notably, it achieves a remarkable $54.44\%$ improvement in inference speed at a resolution of $2048 \times 2048$ over Transformer.

## 1 Introduction

In recent years, the field of generative image models in computer vision has witnessed significant advancements, particularly in class-to-image (Gao et al. (2023); Sun et al. (2024); Sauer et al. (2022)) and text-to-image tasks (Yu et al. (2022; 2023); Bao et al. (2023)). Traditional autoregressive generative models, such as VQGAN (Esser et al. (2021)) and LlamaGen (Sun et al. (2024)), demonstrate excellent performance in class-conditional generation. In the realm of text-conditional generation, models like Parti (Yu et al. (2021; 2022)) and DALL-E (Ramesh et al. (2021)) convert images into discrete tokens using the image tokenizer and project the encoded text features to caption embeddings via an additional MLP (Chen et al. (2023)), operating in an autoregressive manner for both training and inference. Concurrently, non-autoregressive methods, including MAGE (Li et al. (2023)) and MUSE (Chang et al. (2023)), leverage Masked Image Modeling, transforming images into discrete tokens during training and predicting randomly masked tokens.

Another prominent approach to image generation involves diffusion models (Song & Ermon (2019); Song et al. (2020); Ho et al. (2020); Dhariwal & Nichol (2021); Nichol et al. (2021)), such as LDM (Rombach et al. (2022)) with an UNet backbone. Although these models demonstrate high generation quality, their convolutional neural network architecture imposes constraints that hinder scalability. To address this challenge, Transformer-based generative models, such as DiT (Peebles & Xie (2023)), enhance global modeling capabilities through attention mechanisms and significantly improve generation quality. However, the computational complexity of attention mechanisms increases quadratically with sequence length, which constrains both training and inference efficiency.

Mamba (Gu & Dao (2023)) presents a state-space model (Gu et al. (2022; 2021)) characterized by linear time complexity, offering substantial advantages in managing long sequence tasks. Contemporary image generation efforts, including DiM (Teng et al. (2024)), ZigMa (Hu et al. (2024)), and diffuSSM (Yan et al. (2024)), primarily replace the original Transformer block with a Mamba module. These models enhance both efficiency and scalability. Nevertheless, generating images based on diffusion models typically requires hundreds of iterations, which can be prohibitively time-consuming.

Figure 1: Examples of class-conditional (top) and text-conditional (bottom) image generation using MaskMamba-XL.

To eliminate the quadratic complexity growth with sequence length in Transformer models and the excessive generation iterations in autoregressive models, we introduce MaskMamba that integrates Mamba and Transformer architectures and utilizes non-autoregressive Masked Image Modeling (Ni et al. (2024); Lezama et al. (2022)) for image synthesis. We meticulously redesign Bi-Mamba (Mo & Tian (2024); Zhu et al. (2024)) to render it suitable for masked image generation by replacing the causal convolution with standard convolution. Meanwhile, we select concatenation instead of multiplication in the final stage of Bi-Mamba to reduce computational complexity, notably improving the inference speed by $17.77\%$ compared to Bi-Mamba (Zhu et al. (2024)).

We further investigate various MaskMamba hybrid schemes, including serial and grouped parallel schemes (Shaker et al. (2024)). In serial schemes, we explore alternating layer-by-layer arrangements, as well as placing the Transformer in the last $N/2$ layers. For grouped parallel schemes, we assess the effects of partitioning the model into two or four groups along the channel dimension. Our findings indicate that placing the Transformer in the final layers significantly enhances the model's ability to capture global context. Additionally, we implement an in-context condition that allows our model to perform both class-to-image and text-to-image generation tasks within a single framework as show in Fig.1. Meanwhile, we investigate the placement of condition embeddings (Zhu et al. (2024)) by inserting them at different positions of the input sequence including head, middle, and tail. The results indicate that placing condition embedding at the middle yields optimal performance.

In the experimental section, we substantiate the generative capabilities of MaskMamba through two distinct tasks: class-conditional generation and text-conditional generation, utilizing various model sizes for each task. For class-to-image generation task, we execute training over 300 epochs on the ImageNet1k (Deng et al. (2009)) dataset, benchmarking our MaskMamba against Transformer-based and Mamba-based models of analogous size. The results demonstrate that our MaskMamba outperforms both counterparts with respect to generation quality and inference speed. Furthermore, we train and evaluate on CC3M (Sharma et al. (2018)) dataset, attaining superior performance on CC3M and MS-COCO (Lin et al. (2014)) valid datasets.

In summary, our contributions include:

1. We redesign Bi-Mamba to improve its suitability for masked image generation tasks by replacing causal convolution with standard convolution. Additionally, we substitute multiplication with concatenation at the final stage, resulting in a significant performance boost and a $17.77\%$ increase in inference speed compared to Bi-Mamba.

2. We introduce MaskMamba, a unified generative model that integrates redesigned Bi-Mamba and Transformer layers, enabling class-to-image and text-to-image generation tasks to be performed in the same model through an in-context condition.

3. Our MaskMamba surpasses both Transformer-based and Mamba-based models in terms of generation quality and inference speed on the ImageNet1k and CC3M datasets.

## 2 RELATED WORK

**Image Generation.** The domain of image generation is witnessing significant advancements in current research. Initial autoregressive image generative models (Yu et al. (2021); Ding et al. (2021)),

such as VQGAN (Esser et al. (2021)) and LlamaGen (Sun et al. (2024)), have illustrated the potential to generate high-fidelity images by transforming images into discrete tokens and applying autoregressive models to generate image tokens. The advent of text-to-image generative models, like Parti (Yu et al. (2021; 2022)) and DALL-E (Ramesh et al. (2021)), have further propelled progress within this area. Nonetheless, these models exhibit specific inefficiencies in their generation process. To address these issues, non-autoregressive generative models such as MaskGIT (Chang et al. (2022)), MAGE (Li et al. (2023)), and MUSE (Chang et al. (2023)) enhance generation efficiency through Masked Image Modeling. Simultaneously, diffusion models (Song & Ermon (2019); Song et al. (2020); Ho et al. (2020); Dhariwal & Nichol (2021); Nichol et al. (2021); Saharia et al. (2022)), represented by LDM (Rombach et al. (2022)), excel in generation quality despite experiencing scalability constraints linked to their convolutional neural network-based architecture. To overcome these limitations, Transformer-based generative models, including DiT (Peebles & Xie (2023)), advance global modeling capabilities by incorporating attention mechanisms. However, these models continue to struggle with the quadratic increase in computational complexity when processing extensive sequences.

**Mamba Vision.** The Transformer (Vaswani (2017)), established as a leading network architecture, is extensively utilized across various tasks. Nonetheless, its quadratic computational complexity presents significant obstacles for the efficient handling of long sequence tasks. In recent developments, the advent of a novel State-Space Model (Gu et al. (2021)), denominated as Mamba (Gu & Dao (2023); Dao & Gu (2024)), has shown substantial promise in tackling long sequence tasks, capturing significant interest within the research community. The Mamba architecture has effectively supplanted conventional Transformer frameworks in multiple domains, delivering noteworthy results. The Mamba family (Gao et al. (2024); Hatamizadeh & Kautz (2024); Lieber et al. (2024); Pilault et al. (2024)) encompasses a broad spectrum of applications, including text generation, object recognition, 3D point cloud processing, recommendation systems, and image generation, with numerous implementations based on frameworks such as Vision-Mamba (Zhu et al. (2024)), U-Mamba (Ma et al. (2024)), and Rec-Mamba (Yang et al. (2024)). Vision-Mamba employs a bidirectional state-space model structure in conjunction with a hybrid Transformer (Hatamizadeh & Kautz (2024)). However, Mamba has not yet been explored in the context of non-autoregressive image generation. Presently, the majority of Mamba-based generative tasks adhere to the diffusion model paradigm (Hu et al. (2024); Teng et al. (2024)), which entails complexities related to training and the number of inference iterations. Addressing these challenges, we have designed a novel hybrid Mamba structure aimed at extending the application of Mamba in non-autoregressive image generation (Li et al. (2023); Chang et al. (2022; 2023)) tasks, integrating it with Masked Image Modeling (He et al. (2022)) for both training and inference, thereby enhancing the efficiency of these processes.

## 3 METHOD

### 3.1 MASKMAMBA MODEL: OVERVIEW

**Overview.** As illustrated in Fig.2, our MaskMamba fundamentally consists of three components. Firstly, the image pixels $x \in \mathbb{R}^{H \times W \times 3}$ are quantized into discrete tokens $q \in \mathbb{Q}^{h \times w}$ via an image tokenizer (Yu et al. (2021); Van Den Oord et al. (2017); Esser et al. (2021)), where $h = H/r$, $w = W/r$, and $r$ represents the downsample ratio of the image tokenizer. These discrete tokens $q \in \mathbb{Q}^{h \times w}$ serve as indices of the image codebook. Then, we randomly sample the masking ratio $m_r$ (range from 0.55 to 1.0), and mask out $m_r \cdot (h \cdot w)$ tokens, replacing them with a learnable mask token $[M]$. Secondly, we transform the class id into a learnable label embedding (Peebles & Xie (2023); Esser et al. (2021)), denoted as $\{cls\}$. On the other hand, regarding the text conditions, we first extract features using a T5-Large Encoder (Colin (2020)) and then map the extracted features to caption embeddings (Chen et al. (2023)), denoted as $\{t_1, t_2, \ldots, t_N\}$. Lastly, we concat condition embeddings $\{cond\}$ with the image token embeddings $\{q_1, q_2, \ldots, q_{h \cdot w}\}$ at middle, where $\{cond\}$ represents $\{cls\}$ or $\{t_1, t_2, \ldots\}$, and add positional embedding to these $\{q_1, q_2, \ldots, [M], \ldots, q_i, cond, q_j, [M], \ldots, q_{h \cdot w}\}$. The training objective is to predict the token indices of the masked regions utilizing cross-entropy loss (Zhang & Sabuncu (2018)).

**Model Configuration.** We present two types of image generation models: class-conditional and text-conditional models. In accordance with the standards established by prior work (Radford et al.

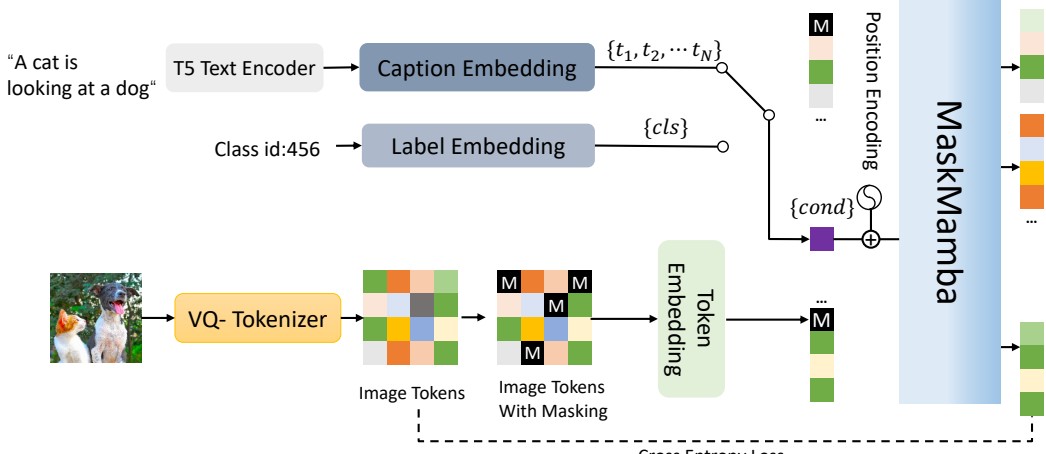

Figure 2: **MaskMamba Pipeline Overview.**

| Type | Model | Params | Layers | Hidden dim |
|------|-------|--------|--------|------------|
|      | MaskMamba-B | 103M | 12 | 768 |
| C2I  | MaskMamba-L | 329M | 24 | 1024 |
|      | MaskMamba-XL | 741M | 36 | 1280 |
| T2I  | MaskMamba-XL | 742M | 36 | 1280 |

Table 1: **Model sizes and configurations of MaskMamba.**

(2019); Touvron et al. (2023)), we adhere to the standard configurations for the Mamba. As shown in Tab.1, we provide three different versions of the class-conditional model, with parameter sizes ranging from 103M to 741M. The generated images have a resolution of $256 \times 256$, and after a downsampling factor of 16, the length of the image token embeddings is set to 256. The length of the class-condition embedding is set to 1, and the length of the text-condition embedding $N$ is set to 120.

## 3.2 MASKMAMBA MODEL: ARCHITECTURE

### 3.2.1 BI-MAMBA-V2 LAYER.

**Convolution Replacement.** As illustrated in Fig.3 (c), we redesign the original Bi-Mamba (Zhu et al. (2024)) architecture to better accommodate tasks associated with masked image generation. We substitute the original causal convolution with a standard convolution. Given the non-autoregressive nature of masked image generation task, the causal convolution only permits unidirectional token mixing, which hinders the potential of non-autoregressive image generation. In contrast, the standard convolution enables tokens to interact bidirectionally across all positions in the input sequence, effectively capturing the global context.

**Symmetric SSM Branch Design.** We incorporate a symmetric SSM branch to better accommodate masked image generation. In the symmetric branch, we first flip the input $x$ before the Backward SSM, then flip it back after the Backward SSM to amalgamate it with the results of the Forward SSM. Additionally, compared to the right-side branch of Bi-Mamba, we employ an extra convolution to mitigate feature loss. To fully exploit the advantages of all the branches, we project the input into a feature space of size $C/2$, thereby ensuring that the final concatenated dimensions are consistent. Our output can be denoted as $X_{out}$, which is computed using the following Eq.1.

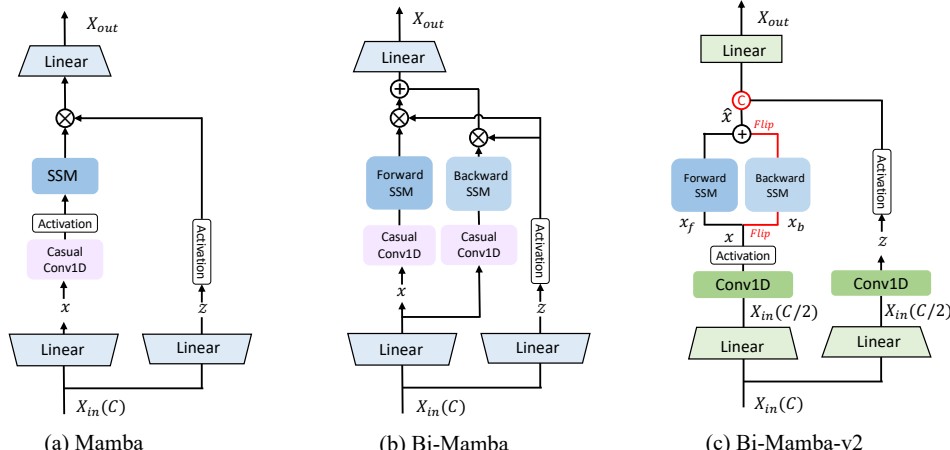

(a) Mamba      (b) Bi-Mamba      (c) Bi-Mamba-v2

Figure 3: (a) Structure of the original Mamba (Gu & Dao (2023)). (b) Bi-Mamba structure proposed in VisionMamba (Zhu et al. (2024)), which introduces a new branch specifically designed for vision tasks. (c) Our redesigned Mamba for masked image generation tasks by using standard convolution instead of causal convolution and replacing the final-stage multiplication with concatenation to reduce computation.

$$
\begin{aligned}
x &= \sigma\left(\mathrm{Conv}\left(\mathrm{Linear}\left(C, C/2\right)\left(X_{\mathrm{in}}\right)\right)\right) \\
x_f &= x, \quad x_b = \mathrm{Flip}(x) \\
\hat{x} &= \mathrm{ForwardSSM}\left(x_f\right) + \mathrm{Flip}\left(\mathrm{BackwardSSM}\left(x_b\right)\right) \\
z &= \sigma\left(\mathrm{Conv}\left(\mathrm{Linear}\left(C, C/2\right)\left(X_{\mathrm{in}}\right)\right)\right) \\
X_{\mathrm{out}} &= \mathrm{Linear}\left(C, C\right)\left(\mathrm{Concat}\left(\hat{x}, z\right)\right)
\end{aligned}
\tag{1}
$$

### 3.2.2 MASKMABA HYBRID SCHEME.

**Group Scheme Design.** As displayed in Fig.4 (a) and Fig.4 (b), we design two group mixing schemes. In group scheme v1, the input is divided into two groups along the channel dimension, which are then processed separately by our Bi-Mamba-v2 layer and Transformer layer. The processed results are then concatenated along the channel dimension and finally feed them into the Norm and Project layers. In group scheme v2, the input is divided into four groups along the channel dimension. Two of these groups are processed by our Bi-Mamba-v2 layer in the Forward SSM and the Backward SSM, while the other two groups are processed by the Transformer layer.

**Serial Scheme Design.** As shown in Fig.4 (c) and Fig.4 (d), we also design two serial mixing schemes. In serial scheme v1, we alternate layer-by-layer arrangements of our Bi-Mamba-v2 and Transformer. In serial scheme v2, we place our Bi-Mamba-v2 in the first $N/2$ layers and Transformer in the last $N/2$ layers. Due to the attention mechanism of Transformer, which can better enhance feature representation, Transformer layer are placed after Mamba layer in all serial modes.

### 3.3 IMAGE GENERATION BY MASKMAMBA

We employ masked image generation methods (Li et al. (2023); Chang et al. (2022)) for image synthesis. During the forward pass, we first initialize 256 masked tokens for generating a resolution of $256 \times 256$ image. Subsequently, we concatenated the condition embeddings with mask tokens at the middle position. Inspired by the iterative generation approach of MUSE (Chang et al. (2023)), our decoding process also adopts a cosine schedule (Chang et al. (2022)) that chooses a fixed proportion of the highest-confidence masked tokens for prediction at each step. These tokens are then set unmasked for remaining steps and the set of masked tokens is correspondingly reduced. Through this methodology, we can infer 256 tokens utilizing merely 20 decoding steps, in contrast to the 256 steps necessitated by autoregressive methods (Touvron et al. (2023); Sun et al. (2024)).

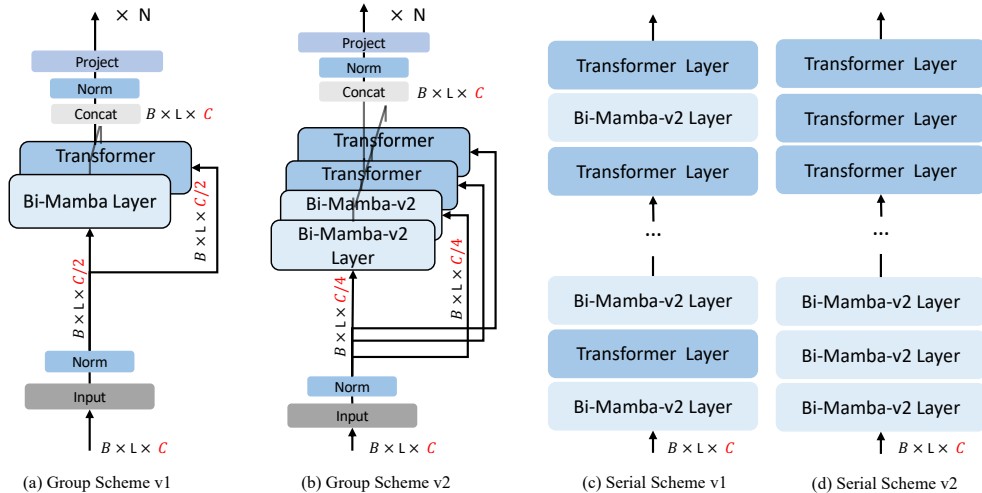

Figure 4: We design two categories of hybrid configurations: group parallel and cascading serial. In group scheme, the model is divided into two or four groups. In serial scheme, we use a layer-wise interleaved structure of Bi-Mamba-v2 and Transformer, or place Bi-Mamba-v2 in the first $N/2$ layers and Transformer in the last $N/2$ layers.

**Class-conditional image generation.** The label embeddings are derived from the index of each category. These label embeddings are concatenated with the masked tokens and MaskMamba gradually predicts these mask tokens through a cosine schedule.

**Text-conditional image generation.** We first extract text features using a T5-Large Encoder (Colin (2020)) and then transform the extracted features to caption embeddings. Similar to the label embeddings, we concatenate these caption embeddings with the masked token embeddings. MaskMamba gradually predicts these mask tokens through a cosine schedule.

**Classifier-free guidance image generation.** The classifier-free guidance (CFG) method proposed by diffusion models (Ho & Salimans (2022)) is a highly effective technique for enhancing the conditional generation capabilities of models, particularly in handling text and image features. Thus, we apply this approach to our model. During the training phase, to simulate the process of unconditional image generation, we randomly drop the condition embeddings with a probability of 0.1. In the inference phase, the logit $\ell_g$ for each token is determined by the following equation: $\ell_g = (1-s)\ell_u + s(\ell_c)$, where $\ell_u$ is uncondition logit, $\ell_c$ is condition logit and $s$ is scale of the CFG.

## 4 EXPERIMENTAL RESULTS

### 4.1 CLASS-CONDITIONAL IMAGE GENERATION

**Training Setup.** All of class-to-image generation models are trained for 300 epochs on the ImageNet $256 \times 256$ dataset, with consistent training parameter settings across all models. Specifically, the base learning rate is set to 1e-4 per 256 batch size, and the global batch size is 1024. Additionally, we employ the AdamW optimizer with $\beta_1 = 0.9$ and $\beta_2 = 0.95$. The dropout rate is consistently set to, including for conditions. During training, the mask rate varies from 0.5 to 1. All training and inference of the models are conducted on V100 GPUs with 32GB of memory.

**Evaluation Metrics.** We use FID-50K (Heusel et al. (2017)) as the primary evaluation metric, while employing Inception Score (Salimans et al. (2016)) (IS) and Inception Score standard deviation (IS-std) as assessment criteria. On the ImageNet validation dataset, we generate 50,000 images based on the CFG and evaluate all models using the aforementioned metrics.

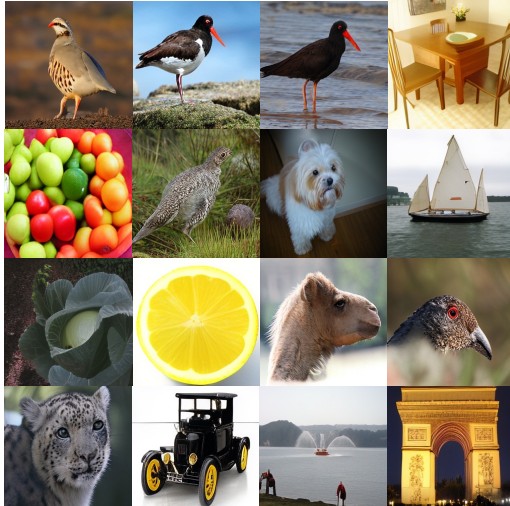

(a) MaskMamba-L with cfg=3.0,iterations=20     (b) MaskMamba-XL with cfg=3.0,iterations=20

Figure 5: Examples of class-conditional image generation using MaskMamba-L (left) and MaskMamba-XL (right) with cfg=3.0, iterations=25.

| Type | Model | Parameters | FID-50k↓ | IS↑ | Precision↑ | Recall↑ | Steps |
|------|-------|------------|----------|-----|-----------|---------|-------|
| AR | VQGAN (Esser et al. (2021)) | 227M | 18.64 | 80.4 | 0.78 | 0.26 | 256 |
| | VQGAN (Esser et al. (2021)) | 1.4B | 15.78 | 74.3 | - | - | 256 |
| | LlamaGen-B (Sun et al. (2024)) | 111M | 8.69 | 124.33 | 0.78 | 0.46 | 256 |
| Mask | MAGE-B (Li et al. (2023)) | 200M | 11.10 | 81.17 | - | - | 20 |
| | MAGE-L (Li et al. (2023)) | 463M | 9.10 | 105.1 | - | - | 20 |
| | MaskGIT (Chang et al. (2022)) | 227M | 6.18 | **182.1** | **0.83** | 0.57 | 10 |
| | Transformer-B | 101M | 11.72 | 90.11 | 0.73 | 0.50 | 25 |
| | Transformer-L | 324M | 7.08 | 127.44 | 0.76 | 0.55 | 25 |
| | Transformer-XL | 736M | 5.96 | 140.81 | 0.75 | 0.58 | 25 |
| | MaskMamba-B | 103M | 10.88 | 89.84 | 0.70 | 0.55 | 25 |
| | MaskMamba-L | 329M | 6.61 | 127.74 | 0.73 | 0.59 | 25 |
| | MaskMamba-XL | 741M | **5.79** | 139.30 | 0.73 | **0.60** | 25 |

Table 2: **Model comparisons on class-conditional Generation on ImageNet** $256 \times 256$ **benchmark.** We utilized FID-50K as the primary evaluation metric, supplemented by Inception Score(IS) as an auxiliary assessment criterion. During the generation process, with cfg set to 3.0.

### 4.1.1 QUALITATIVE RESULTS

**Comparisons with Other Image Generation Methods.** As shown in Tab.2, we compare our MaskMamba model with popular image generation models, including autoregressive (AR) methods (Esser et al. (2021); Sun et al. (2024)), mask-prediction models (Mask) (Li et al. (2023); Chang et al. (2022)), and Transformer-based models (Masked Image Modeling training with the same hyperparameters), focusing on the differences in their backbone networks. MaskMamba adopts the serial scheme v2 mode. Comparisons across various model sizes show that MaskMamba exhibits competitive performance. As illustrated in Fig.5, we randomly select images from MaskMamba-XL models demonstrate high-quality results even when trained only on ImageNet.

### 4.1.2 EXPERIMENT ANALYSIS

**Effect of Class-Free Guidance(CFG) and generation interations.** Fig.6 (a) shows FID and IS variations with the number of iterations in image generation with cfg set to 3. The model achieves

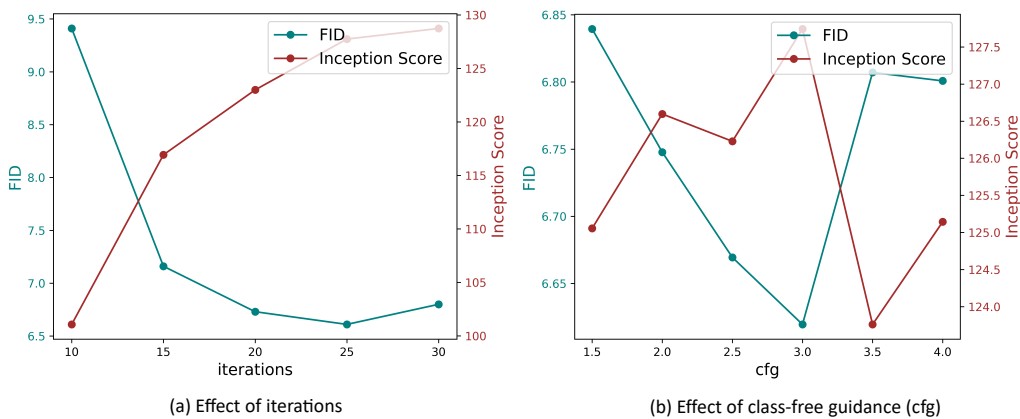

(a) Effect of iterations

(b) Effect of class-free guidance (cfg)

Figure 6: (a) The variation of FID and IS with respect to the number of generation iterations. (b) The scores of FID and IS under different cfg settings.

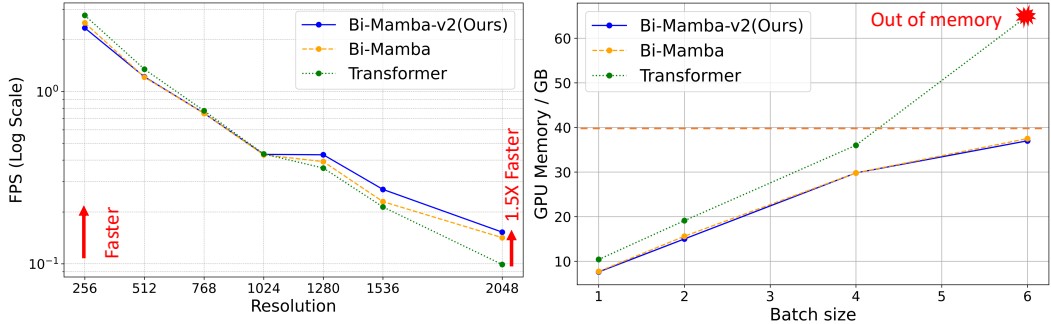

Figure 7: The comparison of inference speed (left) and GPU memory (right) for the Bi-Mamba-v2, Bi-Mamba, and Transformer. Infer speed varies by resolution, while GPU memory varies by batch size.

best performance at 25 iterations and further increasing iterations would deteriorate FID. Fig.6 (b) shows FID and IS scores for different cfg settings, indicating that while class-free guidance enhances visual quality and the model achieves best performance at cfg=3.

**Efficency Analysis.** We conduct a series of experiments to evaluate the effectiveness of our re-designed Bi-Mamba-v2 layer, the original Bi-Mamba layer, and the Transformer layer. To assess inference experiments on higher-resolution images, we primarily focus on inference speed and memory usage with a single layer. All efficiency analysis are conducted on an A100 40G device, comparing the inference speed of these models at different resolutions, as shown in Fig. 7. The results indicate that when the resolution is below $1024 \times 1024$, our Bi-Mamba-v2 layer and the Bi-Mamba layer are slightly slower than the Transformer layer. However, when the resolution exceeds $1024 \times 1024$, our Bi-Mamba-v2 layer is faster than both the Transformer and Bi-Mamba layer. Notably, at a resolution of $2048 \times 2048$, our Bi-Mamba-v2 layer is 1.5 times faster than the Transformer layer. We also compare GPU memory usage at different batch sizes. The memory usage of our Bi-Mamba-v2 layer is comparable to that of the Bi-Mamba layer, while the Transformer layer, due to its quadratic complexity, exhibits a rapid increase in memory usage as the batch size increases. When the batch size reaches 6, the Transformer layer consumes 63GB of GPU memory, leading to out of memory, while our Bi-Mamba-v2 layer requires only 38GB. These experimental results demonstrate that our Bi-Mamba-v2 Layer can generate images at a faster speed and with lower memory usage.

**Effect of different hybrid schemes.** As indicated in Tab.3, we perform a comparative analysis of the image generation outcomes under various hybrid configurations of MaskMamba, categorized into two types: parallel and serial. As depicted in Fig.4, in the grouped parallel configurations, we examine the effects of dividing the model into two and four groups. In the layered serial config-

| Model | Scheme | Parameters | FID-50k ↓ | IS ↑ |
|---|---|---|---|---|
| MaskMamba-L-Group-v1 | - | 327M | 10.04 | 96.35 |
| MaskMamba-L-Group-v2 | - | 278M | 8.95 | 102.72 |
| MaskMamba-L-Serial-v1 | MSMS...MSMS | 329M | 7.45 | 115.90 |
| MaskMamba-L-Serial-v2 | MMMM...SSSS | 329M | **6.73** | **122.99** |

Table 3: **Comparison of different hybrid schemes in MaskMamba for class-conditional Generation on ImageNet** $256 \times 256$ **benchmark.** For all models, the cfg scale is 3 and the number of iteration is 20.

| Backbone | Parameters | FID-50k ↓ | IS ↑ |
|---|---|---|---|
| Bi-Mamba-L | 377M | 12.29 | 87.39 |
| ***Bi-Mamba-V2-L*** | 333M | 8.97 | 103.97 |
| Transformer-L | 324M | 7.08 | 127.44 |
| (Bi-Mamba + Transformer)-L | 358M | 7.82 | 110.87 |
| (***Bi-Mamba-V2*** + Transformer)-L | 329M | **6.61** | **127.74** |

Table 4: **Comparison of different backbones in MaskMamba for class-conditional Generation on ImageNet** $256 \times 256$ **benchmark.** For all models, the cfg scale is 3 and the number of iteration is 25.

urations, we design an interleaved structure of Bi-Mamba-v2 and Transformer {MSMS...MSMS}, as well as an alternative configuration {MMMM...SSSS} where the first $N/2$ layers are Mamba and the subsequent $N/2$ layers are Transformer. The findings from these experiments elucidate the performance and efficiency of the different hybrid configurations.

**Effect of different backbones.** We conduct ablation experiments on different backbones: Bi-Mamba proposed in VisionMamba (Zhu et al. (2024)), redesigned ***Bi-Mamba-V2***, and Transformer (Vaswani (2017)). Bi-Mamba-L uses only the original Bi-Mamba as a layer, while ***Bi-Mamba-V2-L*** uses redesigned Bi-Mamba-v2. The Transformer consists solely of the Transformer architecture. In (Bi-Mamba + Transformer)-L, the first $N/2$ layers are Bi-Mamba, followed by $N/2$ layers of the Transformer. In (***Bi-Mamba-V2*** + Transformer)-L, the first $N/2$ layers are Bi-Mamba-v2, followed by $N/2$ layers of the Transformer. The results indicate that the redesigned Bi-Mamba-v2 outperforms the original Bi-Mamba, and the combination of Mamba with Transformer yields even better performance. Consequently, we select (***Bi-Mamba-V2*** + Transformer) for MaskMamba.

**Effect of different condition positions.** We conduct ablation experiments to assess the impact of the placement of condition embedding $cond$ on model performance. Specifically, we examine the effects of concatenating the condition embedding at different positions in the sequence, such as the head, middle, and tail. The experimental results indicate that optimal performance is achieved when the condition embedding is placed in the middle. This outcome is primarily attributed to the mechanism of selective scan. Since we randomly mask a portion of the image tokens, positioning the condition embedding at either the beginning or the end leads to insufficient supervisory information for controlling conditional generation, primarily due to the increased attention distance. distance.

| Condition postions | Scheme | FID-50k ↓ | IS ↑ |
|---|---|---|---|
| Head | $\langle cond, q_1, q_2, \ldots, \ldots, q_{h \cdot w} \rangle$ | 7.15 | 117.71 |
| Middle | $\langle q_1, q_2, \ldots, cond, \ldots, q_{h \cdot w} \rangle$ | 6.73 | 122.99 |
| Tail | $\langle q_1, q_2, \ldots, \ldots, q_{h \cdot w}, cond \rangle$ | 7.04 | 119.78 |

Table 5: **Comparison of different condition postions in sequence for class-conditional generation on ImageNet** $256 \times 256$ **benchmark.** For all models, the cfg scale is 3 and the number of iteration is 20.

| Backbone | Parameters | FID-CC3M-10K ↓ | IS ↑ | FID-COCO-30K ↓ | IS ↑ |
|----------|-----------|----------------|------|----------------|------|
| Transformer-XL | 736M | 19.20 | 14.98 | 43.21 | 14.44 |
| MaskMamba-XL | 741M | 18.11 | 16.90 | 25.93 | 18.34 |

Table 6: **Comparison of Transformer-XL and MaskMamba-XL for text-conditional generation on CC3M (Sharma et al. (2018)) and MS-COCO (Lin et al. (2014)) datasets.** For all models, the cfg scale is 3 and the number of iteration is 20.

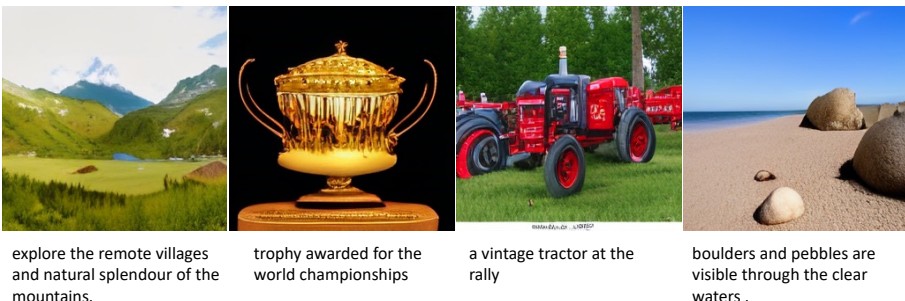

explore the remote villages and natural splendour of the mountains.

trophy awarded for the world championships

a vintage tractor at the rally

boulders and pebbles are visible through the clear waters .

Figure 8: Examples of text-conditional image generation using MaskMamba-XL, with the corresponding captions below the images.

### 4.2 TEXT-CONDITIONAL IMAGE GENERATION

**Training setup.** Similar to the class-conditional training strategy, we adapt a Masked Generative non-autoregressive training strategy for the text. We train the model for 30 epochs on the CC3M (Sharma et al. (2018)) dataset with an image resolution $256 \times 256$. The training parameters are consistent with those of the previous experiments, the base learning rate is set to 1e-4 per 256 batch size, and the global batch size is 1024. Additionally, we employ the AdamW optimizer with $\beta_1 = 0.9$ and $\beta_2 = 0.95$.

**Model trained on CC3M.** As shown in Table 6, we compare the performance of Transformer-XL and our MaskMamba-XL in text-to-image generation, evaluating the FID and IS on the validation sets of CC3M and MS-COCO. Our results consistently outperform the Transformer-based model. As displayed in Figure 8, we utilize text from CC3M as prompts to generate images. MaskMamba-XL is capable of producing high-quality images. However, due to limited training data and the imprecision of the text descriptions in the CC3M dataset, some of generated images exhibit limitations.

## 5 CONCLUSION.

In this work, we propose MaskMamba, a novel hybrid model that integrates Bi-Mamba-v2 and Transformer architectures, utilizing Masked Image Modeling for non-autoregressive image synthesis. We not only redesign a new Bi-Mamba-v2 structure to enhance its suitability for image generation but also investigate the effects of different model mixing strategies and the placement of condition embeddings, ultimately identifying the optimal settings. Additionally, we provide a series of class-conditional image generation models and text-conditional image generation models within a single framework that incorporates an in-context condition. Our experiment results indicate that our MaskMamba surpasses both Transformer-based and Mamba-based models in terms of generation quality and inference speed. We hope our Masked Image Modeling for non-autoregressive image synthesis in MaskMamba will inspire further exploration in Mamba image generation tasks.

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

# A APPENDIX

## A.1 BASE MODEL CONFIGURATIONS

Our MaskMamba Training configurations is given in Tab.7.

| Configuration | Value |
|---|---|
| Optimizer | AdamW |
| Optimizer momentum | $\beta_1 = 0.9, \beta_2 = 0.95$ |
| Base learning rate | 1e-4 |
| Learning rate schedule | cosine decay |
| Training epochs | 300 |
| Warmup epochs | 30 |
| Weight decay | 0.05 |
| EMA | 0.999 |
| Mask ratio min | 0.5 |
| Mask ratio max | 1.0 |

Table 7: Training hyperparameters for MaskMamba.

## A.2 PSEUDO-CODE FOR BI-MAMBA-V2

---
**Algorithm 1** PyTorch-like pseudo-code for Bi-mamba-v2-block
---

```python
class BiMambaBlockV2(nn.Module):
    def __init__(self, dim, mlp_ratio=4.0, drop=0., drop_path=0.,
    act_layer=nn.GELU):
        super().__init__()
        self.attn = Bi-MambaLayer-v2(dim, d_state=16)
        self.attn_flip = Bi-MambaLayer-v2(dim, d_state=16)

        # proj weights
        self.attn_flip.in_proj.weight = self.attn.in_proj.weight
        self.attn_flip.in_proj.bias = self.attn.in_proj.bias
        self.attn_flip.out_proj.weight = self.attn.out_proj.weight
        self.attn_flip.out_proj.bias = self.attn.out_proj.bias

        self.drop_path = DropPath(drop_path) if drop_path > 0. else nn.
    Identity()

        self.norm2 = RMSNorm(dim)
        mlp_hidden_dim = int(dim * mlp_ratio)
        self.mlp = Mlp(in_features=dim, hidden_features=mlp_hidden_dim,
    act_layer=act_layer, drop=drop)

    def forward(self, x):
        # bi mamba add
        x = x + self.drop_path(self.attn(x) + self.attn_flip(x.flip(dims
    =(1,))).flip(dims=(1,)))
        x = x + self.drop_path(self.mlp(self.norm2(x)))
        return x
```

---

## A.3 EXAMPLES OF CLASS-CONDITIONAL IMAGE GENERATION USING MASKMAMBA

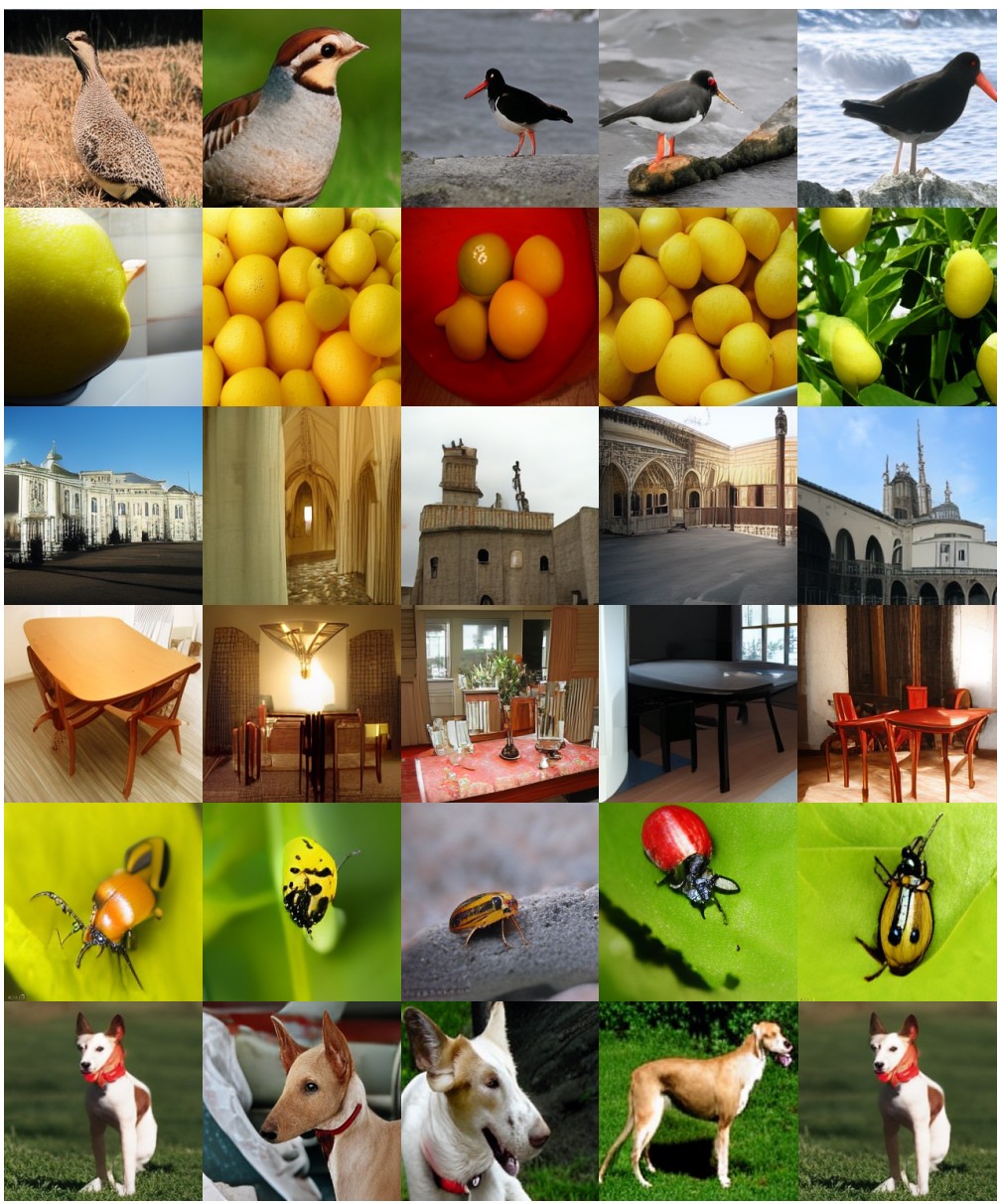

Figure 9: Examples of class-conditional image generation using MaskMamba-B with cfg=3.0, iterations=25.

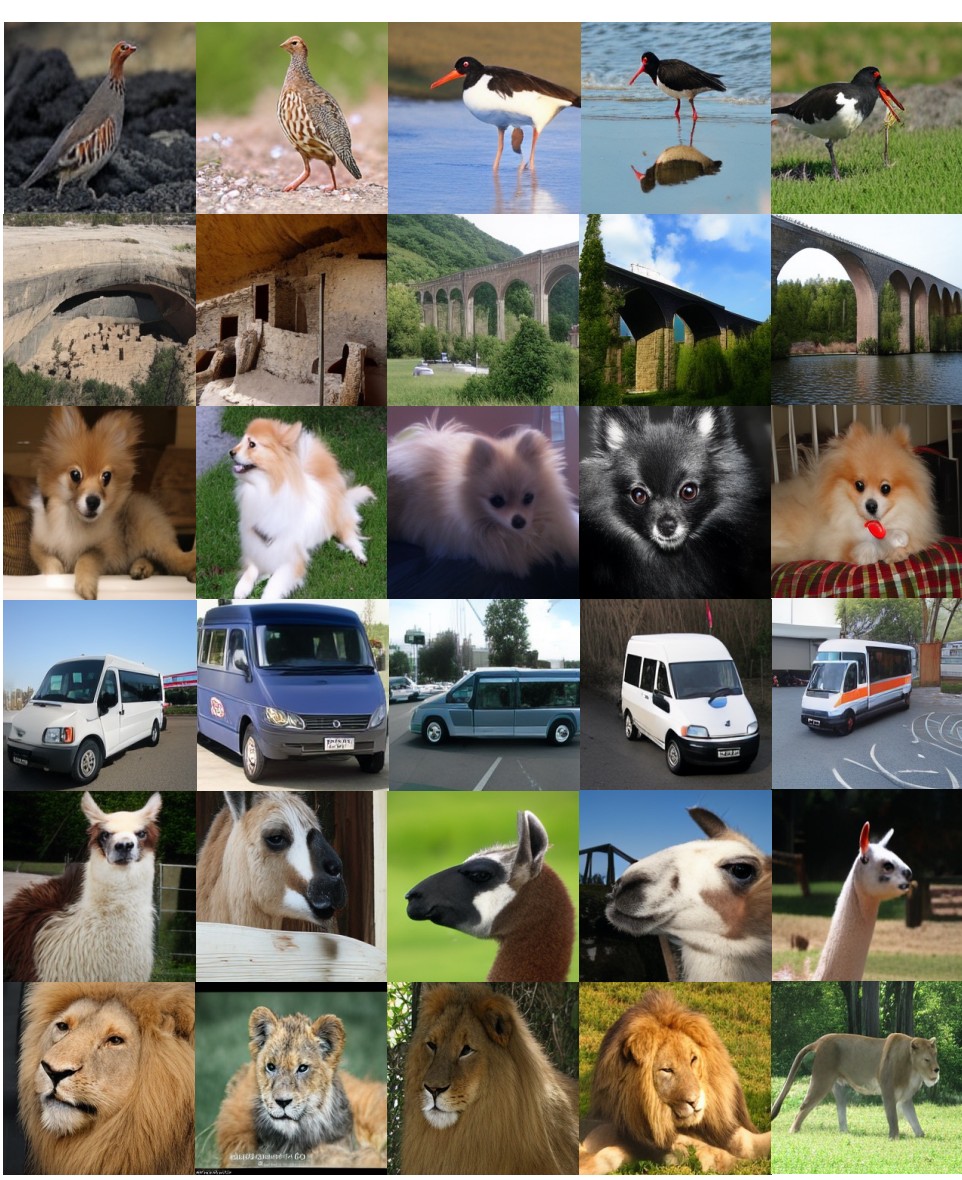

Figure 10: Examples of class-conditional image generation using MaskMamba-L with cfg=3.0, iterations=25.

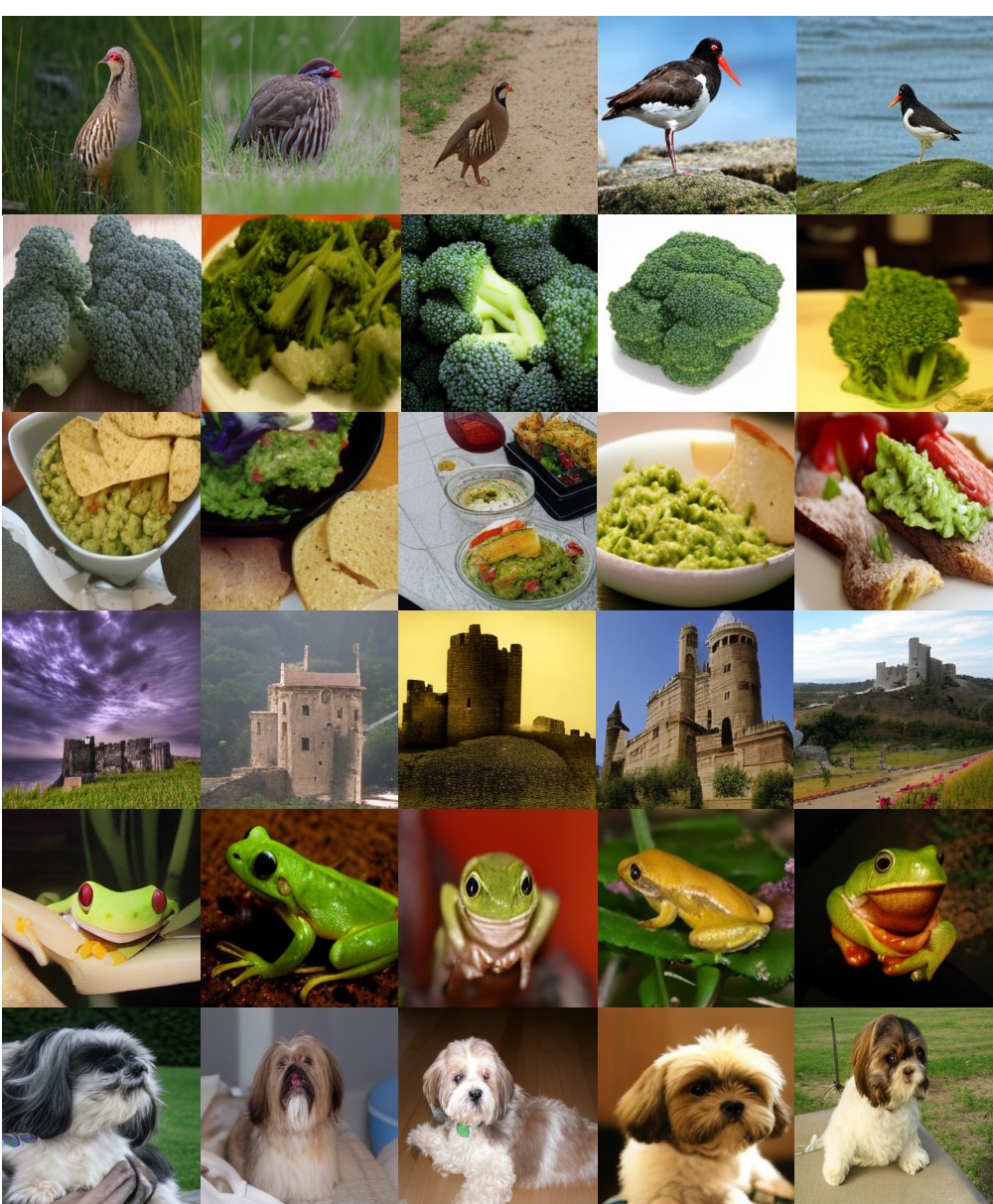

Figure 11: Examples of class-conditional image generation using MaskMamba-XL with cfg=3.0, iterations=25.

