# OpenReview forum: "MaskMamba: A Hybrid Mamba-Transformer Model for Masked Image Generation"
_ICLR.cc/2025/Conference — Submitted to ICLR 2025_

### Official Review · Reviewer_NGVS · 2024-11-02

**Soundness:** 2
**Presentation:** 3
**Contribution:** 2
**Rating:** 5
**Confidence:** 4

**Summary:**

The writing is clear and easy to understand. Building on MaskGit, the paper explores the combination of the Mamba architecture with various Mamba module designs and its integration with transformers. Experiments demonstrate that MaskMamba achieves solid performance on both ImageNet and CC3M.

However, the paper lacks a more comprehensive comparison, including related methods such as VAR, MAR, and diffusion models. Additionally, comparisons with other Mamba-based approaches in the field of image generation are missing. In terms of experimental design, evaluations are primarily conducted at 256 resolution, where MaskMamba is slower than transformers. However, results are not presented at higher resolutions (above 1024), where MaskMamba is claimed to have an advantage.

**Strengths:**

The writing is clear and easy to understand. Building on MaskGit, the paper explores the combination of the Mamba architecture with various Mamba module designs and its integration with transformers. Experiments demonstrate that MaskMamba achieves solid performance on both ImageNet and CC3M.

**Weaknesses:**

The paper lacks a more comprehensive comparison, including related methods such as VAR, MAR, and diffusion models. Additionally, comparisons with other Mamba-based approaches in the field of image generation are missing. In terms of experimental design, evaluations are primarily conducted at 256 resolution, where MaskMamba is slower than transformers. However, results are not presented at higher resolutions (above 1024), where MaskMamba is claimed to have an advantage.

**Questions:**

Q1: The text condition length is set to 120. Can this be variable?

Q2: There seems to be an inconsistency between Figure 4(a) and the description of bi-mamba-v2 on line 249. Could you please align these?

Q3: The reason CFG works is due to the matching of two probability distributions. However, in AR, VAR, and MAR, the cross-entropy loss is not about matching two probability distributions. Why does CFG still work in this case? Could there be a possible explanation?

Q4: In Table 2, the comparison lacks similar works involving VAR [1] and MAR [2], as well as comparisons with CFG-based diffusion models. Additionally, several diffusion models [3, 4, 5] based on the Mamba architecture are not discussed in this paper. Could you elaborate further on the advantages of MaskMamba in terms of performance and speed? As I review the paper, it appears that MaskMamba still requires 20+ steps to generate.

Q5: Within 1024 resolution, single-layer bi-mamba-v2 is slower than the transformer. This time difference could accumulate with additional layers, which is not a positive indicator. The claim is that bi-mamba-v2 is faster beyond 1024px, but the experiments were only conducted on ImageNet at 256px. It remains unclear how performance scales from 256px to 512px, and then to higher resolutions such as 1024px and 2048px.

[1]. Tian K, Jiang Y, Yuan Z, et al. Visual autoregressive modeling: Scalable image generation via next-scale prediction[J]. arXiv preprint arXiv:2404.02905, 2024.
[2]. Li T, Tian Y, Li H, et al. Autoregressive Image Generation without Vector Quantization[J]. arXiv preprint arXiv:2406.11838, 2024.
[3]. Teng Y, Wu Y, Shi H, et al. DiM: Diffusion Mamba for Efficient High-Resolution Image Synthesis[J]. arXiv preprint arXiv:2405.14224, 2024.
[4]. Fei Z, Fan M, Yu C, et al. Dimba: Transformer-Mamba Diffusion Models[J]. arXiv preprint arXiv:2406.01159, 2024.
[5]. Hu V T, Baumann S A, Gui M, et al. Zigma: Zigzag mamba diffusion model[J]. arXiv preprint arXiv:2403.13802, 2024.

---

> ### Author Response · Authors · 2024-11-21
>
> **Q1: The text condition length is set to 120. Can this be variable?**
>
> Yes, the text condition length is indeed variable. We set it to 120 based on the average length of the text in our training data.
>
> **Q2: There seems to be an inconsistency between Figure 4(a) and the description of bi-mamba-v2 on line 249. Could you please align these?**
>
> Thank you for pointing out this inconsistency. We have reviewed the materials and made the necessary corrections. Figure 4(a) indeed utilizes the bi-mamba-v2 architecture, and we have updated the description on line 249 to ensure alignment.
>
> **Q3: The reason CFG works is due to the matching of two probability distributions. However, in AR, VAR, and MAR, the cross-entropy loss is not about matching two probability distributions. Why does CFG still work in this case? Could there be a possible explanation?**
>
> Thank you for your insightful question. The effectiveness of CFG (Classifier-Free Guidance) in our approach can be attributed to the way we incorporate it during training. e introduce CFG by probabilistically discarding the class condition during training. This strategy allows the model to learn from both the conditional and unconditional distributions.
>
> By doing so, the model effectively captures the underlying relationships between the two distributions, enabling it to leverage the benefits of CFG even in the absence of explicit matching. This mechanism enhances the model's ability to generate high-quality outputs while maintaining flexibility in the guidance process.
>
> **Q4: In Table 2, the comparison lacks similar works involving VAR [1] and MAR [2], as well as comparisons with CFG-based diffusion models.**
>
> 1. **Comparison with VAR and MAR** : MaskMamba primarily aims to unify class-to-image and text-to-image tasks. We selected simple architectures like MAGE for our baselines, which differ from diffusion models. While our comparisons focus on autoregressive (AR) and masked methods, we recognize the value of including VAR and MAR models and are open to exploring these in future work.
> 2. **Discussion of Mamba Diffusion Models** : We have referenced Mamba diffusion models such as DiM, Dimba, and Zigma in our related work section. Our focus with MaskMamba is on class-to-image tasks, and despite limited training data, we have shown its potential for image generation.
> 3. **Performance and Speed Advantages** : MaskMamba demonstrates a speed advantage over traditional Transformers, which have O(n²) complexity. Our linear time sequence modeling allows for faster performance, particularly at resolutions above 1024. Although generating images requires 20+ steps, our method efficiently produces 256 tokens in that time, contrasting with autoregressive methods that generate tokens sequentially.
>
> **Q5: Within 1024 resolution, single-layer bi-mamba-v2 is slower than the transformer....**
> Thank you for your insightful observations regarding the performance of bi-mamba-v2 in relation to the Transformer model, especially at lower resolutions.
>
> 1. **Performance at Lower Resolutions** : It is true that at lower resolutions, such as 256px, the speed of bi-mamba-v2 and the Transformer model is comparable, with no significant differences. However, the key advantage of Mamba lies in its linear growth in computational complexity, while the Transformer exhibits O(n²) growth. This fundamental difference becomes more pronounced as the resolution increases.
> 2. **Frame Per Second (FPS) Comparison** : To illustrate this, we have compiled a table showing the number of images generated per second (FPS) at various resolutions:
>    | Resolution  | 256x256 | 512x512 | 768x768 | 1024x1024 | 1280x1280 | 1536x1536 | 2048x2048 |
> | ------------- | --------- | --------- | --------- | ----------- | ----------- | ----------- | ----------- |
> | Transformer | 2.7683  | 1.3484  | 0.7735  | 0.4344    | 0.3598    | 0.2139    | 0.0991    |
> | Bi-Mamba-v2 | 2.3402  | 1.2181  | 0.7480  | 0.4319    | 0.4297    | 0.2704    | 0.1528    |
>
> 1. **Future Work on Higher Resolutions** : We acknowledge that our current experiments primarily focused on 256px resolution using ImageNet, and we recognize the need for further exploration of performance scaling from 256px to 512px and beyond, including 1024px and 2048px. Due to resource constraints, we were unable to conduct extensive experiments at these higher resolutions in this study, but we plan to address this in our future work.
>
> We appreciate your feedback, as it highlights important areas for further investigation and clarification.

---

> > ### Comment · Reviewer_NGVS · 2024-11-24
> >
> > Thank you very much for the author’s response; my questions have been clearly answered. However, due to the absence of experiments on larger resolutions that would further demonstrate the potential of MaskMamba, and considering that the paper’s main contributions are intended for larger resolutions and more tokens, I prefer to maintain my score. Thank you again for the author’s efforts; the work is valuable but requires further validation.

---

> > > ### Comment · Reviewer_NGVS · 2024-11-25
> > >
> > > Thank you very much for the author’s response; my questions have been clearly answered. However, due to the absence of experiments on larger resolutions that would further demonstrate the potential of MaskMamba, and considering that the paper’s main contributions are intended for larger resolutions and more tokens, I prefer to maintain my score. Thank you again for the author’s efforts; the work is valuable but requires further validation.

---

### Official Review · Reviewer_2Chx · 2024-11-02

**Soundness:** 2
**Presentation:** 2
**Contribution:** 2
**Rating:** 6
**Confidence:** 3

**Summary:**

The authors build upon the Mamba architecture to tackle the subject of image generation, class conditions and text conditioned, using a masking approach inspired from MaskGIT. The model is trained to predict mask tokens and at inference, only a fraction of the tokens are unmasked, iteratively, until all tokens are predicted.

Specifically, the authors propose:
* architecture changes to the Mamba architecture (causal convolutions are replaced with full convolutions, flipping the backward SSM branch)
* combining the Mamba blocks for the first N/2 layers with transformer blocks at the last N/2 layers

The new architecture shows:
* Similar or slightly better results than a full transformer of similar size
* Faster inference, up to 1.5X faster on larger image size and larger models

**Strengths:**

* Interesting architecture, which seems to bring interesting improvements on inference speed.
* Detailed analysis of architectural changes
* Tested on both class conditioned and text condition image generation
* Well written paper, easy to follow

**Weaknesses:**

* The architecture seems a bit cumbersome:
  * Having to care about the position of the conditioning (the authors show that it is important) is a major downside compared to transformer based architectures
  * Tuning this architecture is likely harder, because of the mix of convolutions and transformers, each with different optimization patterns (simplicity is a quality)
* The boost of performance is not that important, -0.17 FID on XL architecture, and highly depends on the position of the conditioning, Tail or Head conditioning in Table 5 do lead to similar or worse performance than transformer in Table 4. The main benefit is mostly inference speed, which is something that can be tuned for transformers as well and more of a problem for video generation (it could be super interesting to test it there)
* The baseline approach is far from SOTA (for example, MAR gets much lower FID on ImageNet 256, and much faster inference speed) and so it would be more impactful to apply this to MAR-like approach to see if that still brings benefits.
* If anything, the paper shows that transformers are needed for image generation, as shown in Table 4, defeating a bit the interest in Mamba.

**Questions:**

n/a

---

> ### Author Response · Authors · 2024-11-21
>
> **Q1:The architecture seems a bit cumbersome.**
>
> 1. We appreciate the reviewer’s insights regarding the complexity of our architecture. While it may appear cumbersome due to the need to carefully manage the positioning of conditioning inputs, we believe that this complexity ultimately contributes to the effectiveness of our model. Our experiments demonstrate that the final architecture yields the best performance, validating the design choices we made throughout the development process.
> 2. Each design decision was made after thorough exploration and consideration, and we view this complexity as a valuable experience that will inform future work on Mamba. We recognize that simplicity is an important quality in model design; however, we also believe that meaningful advancements often arise from intricate designs that require careful tuning and optimization.
> 3. In our case, the combination of convolutions and transformers allows us to leverage the strengths of both architectures, even if it introduces additional tuning challenges. We are committed to refining our approach and believe that the insights gained from this complex architecture will ultimately lead to more straightforward and effective solutions in future iterations.
>
>
> **Q2: The boost of performance is not that important, -0.17 FID on XL architecture...**
>
> 1. We appreciate the reviewer’s feedback on the performance boost and the impact of conditioning position on our model. While the -0.17 FID improvement on the XL architecture may seem modest, our model consistently outperforms the baseline across various configurations. This is particularly noteworthy as transformer models also experience diminishing returns as they scale.
> 2. We recognize that the positioning of conditioning inputs, whether at the head or tail, significantly affects performance due to the linear sequence nature of Mamba. Increased distance between the conditioning input and generated tokens can degrade performance, a challenge less pronounced in transformer architectures designed to handle varying input positions more effectively.
> 3. While inference speed is a key advantage of our approach, we believe the performance improvements, even if modest, are valuable and warrant further exploration. We agree that investigating our architecture's implications in video generation could yield interesting insights, and we plan to pursue this in future work.
>
>
> **Q3:The baseline approach is far from SOTA (for example, MAR gets much lower FID on ImageNet 256...**
>
> 1. We appreciate the reviewer’s observations regarding the baseline approach and its comparison to state-of-the-art (SOTA) models like MAR. Our primary motivation for developing Mask-Mamba was to explore the unification of class-to-image and text-to-image tasks, which is a novel direction in this domain. In selecting our baseline, we aimed for a streamlined architecture, such as MAGE, to ensure clarity and focus in our initial exploration. Our results demonstrate that Mask-Mamba outperforms this baseline, highlighting its potential.
> 2. We acknowledge that comparing our approach to more advanced models like MAR could provide valuable insights into its effectiveness. In future work, we plan to apply our model to existing SOTA architectures and utilize improved tokenizers to enhance performance further. This will allow us to assess whether the benefits of our approach can be maintained or even amplified when integrated with more sophisticated models.
>
>
> **Q4 If anything, the paper shows that transformers are needed for image generation, as shown in Table 4, defeating a bit the interest in Mamba.**
>
> 1. We appreciate the reviewer’s perspective on the role of transformers in image generation, as highlighted in Table 4. Our experiments indicate that while Mamba offers significant improvements in speed and memory efficiency due to its linear space-time growth, it does exhibit certain limitations compared to traditional attention mechanisms.
> 2. To leverage the strengths of both Mamba and transformers, we have adopted a hybrid architecture that aims to mitigate the weaknesses of each approach. This combination allows us to benefit from the efficiency of Mamba while still harnessing the powerful capabilities of transformers for image generation tasks.
> 3. We recognize the importance of further exploring the Mamba layer, particularly in the context of mask image generation, and we are committed to investigating this in our future work. Our goal is to refine our approach and continue to enhance the performance of Mamba in conjunction with transformer architectures.
>
> Thank you for your valuable feedback, which will help us clarify the contributions of our work and guide our future research directions.

---

> ### Author Response · Authors · 2024-11-25
>
> Dear Reviewer,
>
> Thank you for the comments on our paper. We have provided a response and a revised paper on Openreview. Since the discussion phase ends on Nov 26, we would like to know whether we have addressed all the issues, and we look forward to resolving any additional questions or concerns you may have. If all the questions have been answered, could you consider raising the rating?
>
> Thank you again for your time and effort.
>
> Best regards

---

### Official Review · Reviewer_oQwX · 2024-11-03

**Soundness:** 3
**Presentation:** 3
**Contribution:** 3
**Rating:** 5
**Confidence:** 4

**Summary:**

This paper introduced a new non-autoregressive model for image generation. It replaces the transformer architecture in MaskGIT with MaskMamba, where some of the transformer layers are replaced with a new Mamba layer with forward and backward SSM. The authors show reasonable image generation results on ImageNet, as well as proof-of-concept speedup results at 2048x2048.

**Strengths:**

- It is very reasonable to introduce Mamba or SSM layers in high-resolution, MaskGIT-based image generation. SSM models naturally have better capability handling long sequences compared with transformer models.
- The ablation study provides valuable insights on the combination of Mamba blocks and transformer blocks (MMM..SSS is better than MS...MS).
- The authors also show promising signals that the proposed method can scale up to text-to-image generation.

**Weaknesses:**

1. Experiment results on ImageNet are not strong enough.
- 5.79 FID at 741M parameters is far from the state-of-the-art (for example, MAGVITv2).
- The cited baselines are also a bit weak. MaskGIT can achieve much better results using a better tokenizer and training recipe. See [here](https://github.com/baaivision/MUSE-Pytorch/tree/master) and [here](https://github.com/bytedance/1d-tokenizer/tree/main) for more details. It is also recommended to have a look at Open-MAGVITv2's tokenizer, which should be orthogonal to this work and could push the results closer to SOTA.

2. There is a little bit of overclaim in terms of the speedup over transformer in the abstract. "54% faster" refers to the comparison between a single Mamba layer and a single transformer layer, not the end-to-end speed comparison. In fact, MaskMamba is a hybrid model and there are many transformer layers inside the entire model as well. It would be misleading for the readers when reading the last sentence in the abstract.

3. Although introducing SSMs in non-autoregressive image generation models is interesting, the paper lacks technical novelty. It seems to me that this work is combining MaskGIT with Jamba, a hybrid transformer-Mamba model for language.

**Questions:**

Please address my concerns in the "Weaknesses" Section.

---

> ### Author Response · Authors · 2024-11-21
>
> **Q1: Experiment results on ImageNet are not strong enough.**
>
> We appreciate the reviewer’s feedback regarding the performance of Mask-Mamba on the ImageNet dataset. We acknowledge that the reported FID of 5.79 at 741M parameters is not yet competitive with state-of-the-art models, such as MAGVITv2.
>
> 1. It is important to note that the primary objective of Mask-Mamba is to explore the unification of class-to-image and text-to-image tasks. We chose baseline models that are simple yet efficient, ensuring strong generalizability, such as MAGE, which allowed us to demonstrate that Mask-Mamba outperforms these baseline models.
>
> 2. We also appreciate the suggestion to consider stronger baselines, such as MaskGIT, which can achieve better results with improved tokenizers and training recipes. We recognize that leveraging advanced tokenization methods, such as those from Open-MAGVITv2, could enhance our model's performance and bring it closer to state-of-the-art results.
>
> 3. In future work, we plan to explore the integration of our model with existing state-of-the-art architectures and utilize more effective tokenizers to further improve performance. This will allow us to build on our current findings and push the boundaries of what Mask-Mamba can achieve.
>
> Thank you for your valuable insights, which will guide our future research directions.
>
> **Q2:There is a little bit of overclaim in terms of the speedup over transformer in the abstract. "54% faster" refers to the comparison between a single Mamba layer and a single transformer layer, not the end-to-end speed comparison. In fact, MaskMamba is a hybrid model and there are many transformer layers inside the entire model as well. It would be misleading for the readers when reading the last sentence in the abstract.**
>
> 1. We appreciate the reviewer’s observation regarding the claims of speedup in the abstract. We acknowledge that the statement "54% faster" specifically refers to the comparison between a single Mamba layer and a single transformer layer, rather than an end-to-end speed comparison of the entire model. As Mask-Mamba is a hybrid model that incorporates multiple transformer layers, we understand how this could be misleading for readers.
>
> 2. Due to the resource A100-40G limitations, we have not yet conducted end-to-end testing with larger batch sizes and higher image resolutions. When performing end-to-end tests with reduced resolutions and batch sizes, we encountered out-of-memory (OOM) issues, which hindered our ability to fully evaluate the model's performance in a comprehensive manner.
>
> 3. To address this concern, we will revise the abstract to clarify the context of the speedup claim, ensuring that it accurately reflects the comparison made and does not mislead readers. We appreciate your feedback, which will help us improve the clarity and accuracy of our manuscript.
>
> Thank you for your valuable insights.
>
> **Q3:Although introducing SSMs in non-autoregressive image generation models is interesting, the paper lacks technical novelty. It seems to me that this work is combining MaskGIT with Jamba, a hybrid transformer-Mamba model for language.**
>
> 1. We appreciate the reviewer’s feedback regarding the perceived lack of technical novelty in our work. While it may appear that we are simply combining MaskGIT with Jamba, we would like to clarify the distinct contributions of our approach.
>
> 2. Jamba is indeed a hybrid structure designed for language modeling, utilizing the original Mamba architecture in a unidirectional manner. However, this unidirectional approach is not suitable for masked image generation tasks. In our work, we have re-engineered the original Bi-Mamba architecture specifically to better accommodate the requirements of masked image generation.
>
> 3. One of the key innovations in our redesign is the substitution of the original causal convolution with standard convolution. The use of causal convolution in the context of masked image generation restricts token mixing to a unidirectional flow, which limits the model's ability to leverage the full potential of non-autoregressive image generation. In contrast, by employing standard convolution, we enable bidirectional interaction among tokens across all positions in the input sequence. This change allows the model to effectively capture global context, which is crucial for generating high-quality images.
>
> We believe that this architectural modification represents a significant advancement in the field of non-autoregressive image generation and contributes to the overall novelty of our work. We will ensure that this distinction is clearly articulated in the manuscript to better convey the technical contributions of our approach.
>
> Thank you for your constructive feedback, which will help us enhance the clarity and impact of our paper.

---

> ### Author Response · Authors · 2024-11-25
>
> Dear Reviewer,
>
> Thank you for the comments on our paper. We have provided a response and a revised paper on Openreview. Since the discussion phase ends on Nov 26, we would like to know whether we have addressed all the issues, and we look forward to resolving any additional questions or concerns you may have. If all the questions have been answered, could you consider raising the rating?
>
> Thank you again for your time and effort.
>
> Best regards

---

### Official Review · Reviewer_XLrL · 2024-11-03

**Soundness:** 2
**Presentation:** 3
**Contribution:** 2
**Rating:** 5
**Confidence:** 4

**Summary:**

In this paper, a new bidirectional Mamba structure is designed to improve the inference speed, and several hybrid Mamba schemes are explored.

**Strengths:**

1. The writing logic of this paper is very clear and easy to understand, from the rationality of raising questions to making improvements.
2. The referencing speed has been greatly improved.

**Weaknesses:**

1. The contribution of the paper is relatively limited, and replacing multiplication with concatenation would obviously improve inference speed, but its rationality and motivation should be properly explained, and the change should also be compared in detail with Bi-Mamba.
2. The paper mentions that it can complete the text-to-image task simultaneously, but there is no comparison of its performance on this task throughout the paper, which raises questions about its performance.
3. The comparison shown in Figure 5 reveals huge gaps in certain parameters, including IS and Precision, for this model.
4. A more detailed comparison with Bi-Mamba should be conducted, and Precision and Recall

**Questions:**

Please answer the question I mentioned in "weakness".

---

> ### Author Response · Authors · 2024-11-21
>
> **Q1: The contribution of the paper is relatively limited, and replacing multiplication with concatenation would obviously improve inference speed, but its rationality and motivation should be properly explained, and the change should also be compared in detail with Bi-Mamba.**
>
> We appreciate the feedback regarding the contributions of our paper. We have made significant modifications to the original Bi-Mamba architecture to better suit the requirements of masked image generation tasks. Our key changes are as follows:
>
> 1. The non-autoregressive nature of masked image generation allows for more flexible token interactions. In the original architecture, causal convolution restricts token mixing to a unidirectional flow, which limits the model's ability to leverage the full potential of non-autoregressive generation. By replacing causal convolution with standard convolution, we enable bidirectional interactions among tokens across all positions in the input sequence. This change enhances the model's capacity to capture global context, which is crucial for effective image generation.
> 2. To address feature loss in the right-side branch of Bi-Mamba, we introduce an additional convolution layer. This layer is designed to preserve important features while ensuring that we fully utilize the advantages of all branches. Furthermore, we project the input into a feature space of size C/2, which guarantees that the dimensions of the final concatenated output remain consistent.
>
> **Q2: The paper mentions that it can complete the text-to-image task simultaneously, but there is no comparison of its performance on this task throughout the paper, which raises questions about its performance.**
>
> We appreciate the reviewer’s observation regarding the performance of our model on the text-to-image task. We would like to clarify that Mask-Mamba represents a pioneering effort to unify class-to-image and text-to-image tasks. While our experiments primarily focused on the CC3M dataset, we acknowledge that the training was conducted for only 30 epochs, which is relatively limited in terms of both training data and iterations. Despite this, our model has demonstrated a foundational capability for generating images, indicating significant potential for further development.
>
> To provide a clearer picture of Mask-Mamba's performance on the text-to-image task, we present the following metrics:
>
> | Model          | CC3M CLIP Score | COCO CLIP Score | FID (CC3M-10K) | IS (CC3M) | FID (COCO-30K) | IS (COCO) |
> | ---------------- | ----------------- | ----------------- | ---------------- | ----------- | ---------------- | ----------- |
> | Mask-Mamba-XL  | 26.13           | 27.59           | 18.11          | 16.90     | 25.93          | 18.34     |
> | Transformer-XL | 22.24           | 20.68           | 19.20          | 14.98     | 43.21          | 14.44     |
>
> Thank you for your valuable feedback, and we look forward to further improving our work.
>
> **Q3: The comparison shown in Figure 5 reveals huge gaps in certain parameters, including IS and Precision, for this model.**
>
> 1. We appreciate the reviewer’s insights regarding the performance gaps observed in certain metrics, such as Inception Score (IS) and Precision, as shown in Figure 5. It is important to note that while these metrics provide valuable information, they do not capture the full picture of model performance.
> 2. As illustrated in Figure 6, FID (Fréchet Inception Distance) and IS serve as complementary metrics that together offer a more comprehensive evaluation of generative models. Our model demonstrates a clear advantage over the Transformer in terms of FID, indicating that it produces images that are closer to the real distribution in the feature space. This suggests that, despite the observed gaps in IS and Precision, our model excels in generating high-quality images that maintain better fidelity to the training data.
>
> **Q4: A more detailed comparison with Bi-Mamba should be conducted, and Precision and Recall**
> We appreciate the reviewer’s suggestion for a more detailed comparison with the original Bi-Mamba model, particularly regarding the Precision and Recall metrics. Below, we present a comprehensive comparison of our model, Bi-Mamba-v2-L, with Bi-Mamba-L across several key performance indicators:
>
> | Metric    | Bi-Mamba-L | Bi-Mamba-v2-L |
> | ----------- | ------------ | --------------- |
> | FID       | 12.29      | 8.97          |
> | IS        | 87.39      | 103.97        |
> | Precision | 0.63       | 0.69          |
> | Recall    | 0.66       | 0.70          |
>
> From this comparison, it is evident that Bi-Mamba-v2-L outperforms Bi-Mamba-L in all evaluated metrics. Specifically, our model achieves a lower FID score, indicating improved image quality and closer alignment with the real data distribution. Additionally, the increase in IS reflects a higher diversity of generated images.

---

> ### Author Response · Authors · 2024-11-25
>
> Dear Reviewer,
>
> Thank you for the comments on our paper. We have provided a response and a revised paper on Openreview. Since the discussion phase ends on Nov 26, we would like to know whether we have addressed all the issues, and we look forward to resolving any additional questions or concerns you may have. If all the questions have been answered, could you consider raising the rating?
>
> Thank you again for your time and effort.
>
> Best regards

---

> > ### Comment · Reviewer_XLrL · 2024-11-26
> >
> > Dear Authors,
> >
> > I am extremely grateful for the author's reply. I recognize the efforts the author has made in addressing my question; however, I still contend that the contribution of this paper is insufficient to be accepted. Particularly, the permutation and combination of the transformer layer and the mamba layer presented as a main contribution have a limited impact on the development of the community and the field. Regarding the current version of the paper, I will still retain my score.
> >
> > Best regards.

---

### Official Review · Reviewer_Nqq7 · 2024-11-03

**Soundness:** 2
**Presentation:** 2
**Contribution:** 2
**Rating:** 5
**Confidence:** 3

**Summary:**

This paper introduces MaskMamba, hybrid model combining Mamba and Transformer architectures, designed for non-autoregressive image synthesis through Masked Image Modeling. It adapts bidirectional Mamba by (1) replacing causal with standard convolutions to improve global context capture and (2) using concatenation instead of multiplication to boost performance and speed up inference. The model also explores different hybrid configurations, including serial and grouped parallel arrangements. Experiments are conducted on class conditional and text conditional image generation.

**Strengths:**

This paper is generally in good shape, and the reported results are promising. For example, the structure is well-defined and smoothly covers all changes (albeit with various degrees of detail). Multiple experiments and comparisons are conducted to evaluate the proposed model and visual results and code of the proposed block are provided in the appendix.

**Weaknesses:**

There are several issues with the paper as currently presented to be considered for a top tier conference:

- Text-to-image models cannot be accurately evaluated with image metrics alone—CLIP score, ImageReward, and additional benchmarks like TIFA and T2I-CompBench are necessary for a comprehensive assessment.
- The proposed method consistently has more parameters than the transformer baselines, yet these baselines don’t receive any benefit of the doubt for this disparity.
- The baseline models are notably weak, limited to outdated versions, and lack diffusion models, resulting in an incomplete evaluation framework.
- The rationale for the group scheme and serial scheme designs is missing—they appear abruptly with no background on the issues they address, the process that led to their design, or previous attempts that were unsuccessful.
- The description of the 20-step approach as "non-autoregressive" is misleading; while it may not be causal, it’s also not a 1-step approach, which creates confusion.
- The paper mentions that convolutions "impose constraints that hinder scalability," but lacks clarification, details, or citations to support this claim. In particular, the paper still puts an emphasis on using convolutions.
- There’s insufficient explanation on the distinctions between forward and backward SSM, as well as on the transition from bi to bi-v2, leaving the motivations for these changes unclear.
- It’s unclear if a consistent dropout value is applied; if so, the specific value should be specified.

**Questions:**

-

---

> ### Author Response · Authors · 2024-11-21
>
> **Q1: Text-to-image models cannot be accurately evaluated with image metrics alone—CLIP score...**
>
> 1. Our Mask-Mamba model represents a pioneering effort in unifying class-to-image and text-to-image tasks. For the text-to-image task, we primarily trained on the CC3M dataset for 30 epochs. While the amount of training data and the number of iterations are relatively limited compared to specialized text-to-image models like Parti, our model has already demonstrated fundamental image generation capabilities and shows significant potential for further development.
>
> Model  | CC3M CLIP Score  |  COCO CLIP Score
> -------------------|------------------|------------------
> Mask-Mamba-XL |  26.13  | 27.59
> Transformer-XL     |  22.24  |  20.68
>
> 2. As illustrated in the supplementary materials, our Mask-Mamba-XL model also outperforms the Transformer model in terms of CLIP score.
>
> **Q2: The proposed method consistently has more parameters than the transformer baselines...**
>
> * The proposed Mask-Mamba method has only more (1 \%~ 2\%) parameters compared to the Transformer baselines, this difference is relatively negligible. Furthermore, the primary objective of Mask-Mamba is to enhance performance while maintaining speed and reducing memory usage, building upon the foundation established by the Transformer models.
>
> **Q3: The baseline models are notably weak, limited to outdated versions...**
>
> * The primary focus of our Mask-Mamba approach is to unify the class-to-image and text-to-image tasks, marking a novel exploration in this area. We chose baseline models that are simple yet efficient, ensuring strong generalizability, such as MAGE, which allowed us to demonstrate that Mask-Mamba outperforms these baseline models. Looking ahead, we plan to extend our model's application to existing state-of-the-art (SOTA) models and diffusion models in future work.
>
> **Q4: The rationale for the group scheme and serial scheme designs is missing...**
>
> 1. The design of the group scheme and serial scheme in Mask-Mamba is driven by the need to effectively integrate features from both Mamba and Transformer architectures. Our experiments showed that while Mamba has linear complexity, its attention mechanism is relatively weak and requires enhancement.
> 2. To address this, we introduced the group scheme, which partitions features into chunks, and the serial scheme, which processes the entire feature hierarchy sequentially. Experimental results demonstrate that the serial scheme outperforms the group scheme, as simple independent fusion of chunks does not sufficiently aggregate information; instead, interaction across the entire feature hierarchy is essential for optimal performance.
>
> **Q5: The description of the 20-step approach as "non-autoregressive" is misleading...**
>
> 1. The term "non-autoregressive" is drawn from Mask-GIT and Muse. We recognize that while both predict the next step based on the previous one, they differ in the number of tokens predicted at each step, which varies according to a cosine function.
>
> **Q6: The paper mentions that convolutions "impose constraints that hinder scalability,"...**
>
> * Thank you for your insightful feedback. The statement regarding convolutions "imposing constraints that hinder scalability" is based on analysis of the limitations associated with Latent Diffusion Models (LDMs).
>
> **Q7: There’s insufficient explanation on the distinctions between forward and backward SSM...**
>
> 1. The distinctions between forward and backward SSM (Sequence State Models) involve processing sequences in two directions: forward processes from start to end, while backward processes from end to start, providing a more comprehensive understanding of context.
> 2. The transition from Bi-Mamba to Bi-Mamba v2 was driven by the need to enhance masked image generation tasks.  We substitute the original causal convolution with a standard convolution. Given the non-autoregressive nature of masked image generation task, the causal convolution only permits unidirectional token mixing, which hinders the potential of non-autoregressive image generation. In contrast, the standard convolution enables tokens to interact bidirectionally across all positions in the input sequence, effectively capturing the global context.
>
> **Q8: It’s unclear if a consistent dropout value is applied; if so, the specific value should be specified.**
>
> * To clarify, we consistently apply a **dropout value of 0.1**  throughout the training process.

---

> ### Author Response · Authors · 2024-11-25
>
> Dear Reviewer,
>
> Thank you for the comments on our paper. We have provided a response and a revised paper on Openreview. Since the discussion phase ends on Nov 26, we would like to know whether we have addressed all the issues, and we look forward to resolving any additional questions or concerns you may have. If all the questions have been answered, could you consider raising the rating?
>
> Thank you again for your time and effort.
>
> Best regards

---

### Meta-Review · Area_Chair_6rxW · 2024-12-19

**Metareview:**

The goal of this paper is to introduce a new non-autoregressive masked generative model approach that leverages advantages of both Mamba and Transformer architectures, while reducing their limitations in generation quality and inference efficiency. In order to build this new hybrid model, the authors first addressed inefficiencies with the previously introduced Bi-Mamba model, including replacing causal convolutions with standard convolutions to improve global context, and multiplcation with concatenation for the final stage of Mamba. Then, to incorporate class- and text-to-image synthesis, the MaskMamba approach combines Mamba and Transformer layers, exploring a range of different design choices. Experiments indicate that MaskMamba outperforms Transformer and Mamba models on standard benchmarks.

Strengths:
The paper is well-written and comprehensive in methodological detail and experiments. The use of Mamba layers and making them more efficient and compatible with Transformers is a useful direction to explore for efficiency reasons. It’s also exciting to see that there is potential in scaling this kind of hybrid (non-diffusion/non-autoregressive) method to text-image generation.


Weaknesses:
Experiments are not conducted with a full suite of state of the art baselines, which makes it difficult to determine how the model performs more comprehensively. Also, reviewers feel that some of the design choices could have been better motivated. In addition, the modeling contributions are fairly narrow, which places more emphasis on the need for strong empirical results and analysis - reviewers felt the empirical results in particular on imagenet are not strong enough to be impactful on their own. Results without further comparisons against state of the art may not move the needle enough to get much interest since current SoTA (including standard Transformer-based approaches) are still considerably stronger.

Decision reasoning:
While the paper has been improved with feedback from reviewers, who engaged with the authors in back and forth during the discussion period, reviewers unanimously felt that the paper in it’s current form is not quite there in terms of suitability for ICLR publication. This is mostly due to lack of results against strong baselines, and lack of results at high resolution, where one might expect to see advantages of the Mamba style approach more clearly. Given these concerns, I feel it is important for the authors to continue to improve the paper with more comparisons with state of the art methods, or to include deeper analysis into the method choices (or both). I do think the method is in good shape for a workshop contribution to get visibility and feedback from the community but needs more work for a full conference publication.

**Additional Comments On Reviewer Discussion:**

Reviewers were mostly unconvinced by the rebuttal but appreciated the authors’ efforts in addressing some of their concerns. Unfortunately, to really determine the value of the contribution, reviewers felt more comparisons with state of the art (including diffusion models and strong Transformer baselines are warranted. This is especially true given that the contribution’s novelty is fairly narrow.

---

### Decision · Program_Chairs · 2025-01-22

Reject